# MARGINAL TAIL-ADAPTIVE NORMALIZING FLOWS

## ABSTRACT

Learning the tail behavior of a distribution is a notoriously difficult problem. The number of samples from the tail is small, and deep generative models, such as normalizing flows, tend to concentrate on learning the body of the distribution. In this paper, we focus on improving the ability of normalizing flows to correctly capture the tail behavior and, thus, form more accurate models. We prove that the marginal tailedness of a triangular flow can be controlled via the tailedness of the marginals of the base distribution of the normalizing flow. This theoretical insight leads us to a novel type of triangular flows based on learnable base distributions and data-driven permutations. Since the proposed flows preserve marginal tailedness, we call them *marginal tail-adaptive flows* (mTAFs). An empirical analysis on synthetic data shows that mTAF improves on the robustness and efficiency of vanilla flows and—motivated by our theory—allows to successfully generate tail samples from the distributions. More generally, our experiments affirm that a careful choice of the base distribution is an effective way to introducing inductive biases to normalizing flows.

## 1 INTRODUCTION

Heavy-tailed distributions are known to occur in various applications in biology, finance, social sciences, and more. Examples for such observations include the length of protein sequences in genomes (Koonin et al., 2006), returns of stocks (Gabaix et al., 2003), or the size of cities (Gabaix, 1999). Applications that are tightly connected to typical deep learning applications include the frequency of class examples in image classification (Horn & Perona, 2017) and the frequency of words (Zipf, 1949) in natural language processing. From a theoretical point of view, this is not surprising since heavy-tailed distributions emerge from several circumstances, including the limiting distribution in the generalized central limit theorem, of a multiplicative process, or as the limit of an extremal process (Nair et al., 2013). Given the frequency of occurrence, developing generative models that allow to learn heavy-tailed distributions is essential.

Normalizing Flows (NFs (Rippel & Adams, 2013; Tabak & Turner, 2013; Dinh et al., 2015; Rezende & Mohamed, 2015)) are a popular class of deep generative models. Despite their success in learning tractable distributions where both sampling and density evaluation can be efficient and exact, their ability to model heavy tailed distributions is known to be limited. Jaini et al. (2020) identified the problem that a range of NFs (e.g. vanilla triangular flows with a Gaussian base distribution) are unable to map a light-tailed distribution to a heavy-tailed distribution. They propose to solve this issue by replacing the Gaussian base distribution by a multivariate $t$-distribution with one learnable degree of freedom. While this allows to model distributions with a heavy-tailed euclidean norm, we show that modeling multivariate distributions, where some of the marginals are heavy- and some are light-tailed, still poses a problem.

**Contributions**   Our contributions in this work, that extend the results of Jaini et al. (2020), are the following. First, we prove that a triangular affine NF using a base distribution with solely heavy-tailed marginals is only able to provide a target distribution with just heavy-tailed marginals as well. Consequently, such a NF is not capable of learning distributions with mixed marginal tail behavior. Second, we derive a result that states conditions under which the marginal tailedness of the base distribution can be preserved. Third, based on these theoretical findings, we propose a novel kind of triangular NF that allows to learn distributions with heavy- and light-tailed marginals. The new model is called *marginally Tail-Adaptive Flows* (mTAFs), and as illustrated in Figure 1, combines

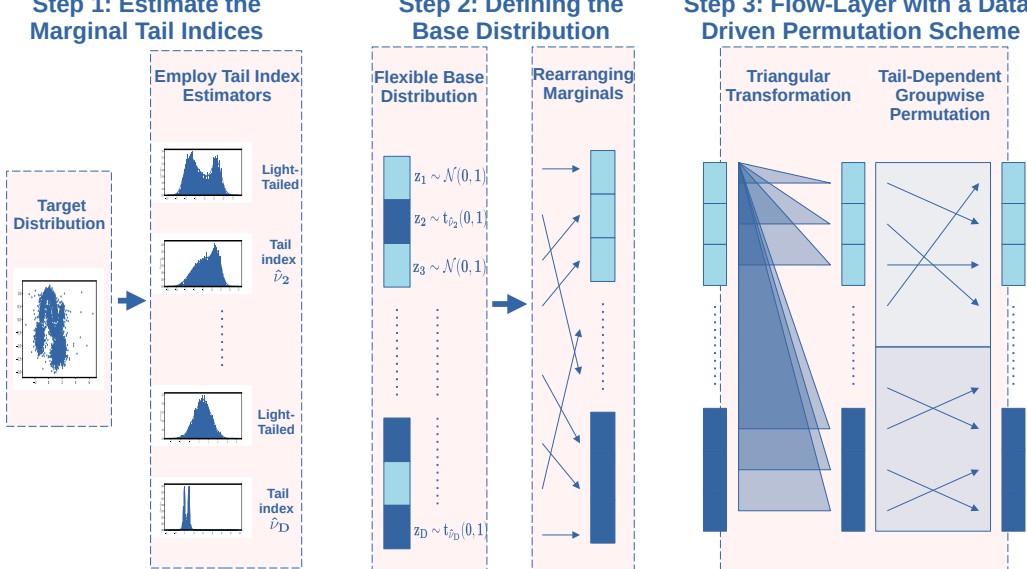

Figure 1: An Overview of mTAF. In a first step, we apply estimators from extreme value theory to classify the marginals as heavy- or light-tailed. This classification defines a flexible base distribution consisting of marginal Gaussians and marginal $t$-distributions with trainable degree of freedom, as illustrated by Step 2 of this figure. Further, we rearrange the marginals such that the first $d$ marginals are light-tailed, whereas the remaining marginals are heavy-tailed. mTAF is then constructed using several flow-layers as visualized in Step 3: we employ a triangular mapping, followed by a 2-group permutation scheme, i.e. we just permute within the set of light-tailed marginals and within the set of heavy-tailed marginals. At the end, we restore the original ordering using the inverse of the permutation employed in Step 2. Using Theorem 3, we prove that mTAFs are marginally tail-adaptive (Corollary 1).

- estimators from extreme value theory (Hill, 1975; Dekkers et al., 1989; Csorgo et al., 1985) to initially assess heavy-tailedness of the target's marginals;

- a flexible and trainable base distribution based on the estimated tail behavior of the target distribution;

- a new permutation-scheme between flow-layers that ensures the correct tail behavior of the estimated target distribution.

Finally, we conduct an experimental analysis demonstrating the superior performance of the proposed mTAFs in comparison to other flow models. Furthermore, we present a new sample generation scheme, motivated by our theory, which successfully generates joint samples that are from the tails of a specified marginal.

**Notational Conventions** In the following, we will denote random variables by bold letters, such as $\mathbf{x}$, and its realisations by non-bold letters, $x$. We use this notation for multivariate and for univariate random variables. Further, we denote the $j$th component of $\mathbf{x}$ by $\mathbf{x}_j$, and $\mathbf{x}_{\leq j}$ or $\mathbf{x}_{<j}$ are the first $j$ or $j-1$ components of $\mathbf{x}$, respectively. We denote the random variable representing the base distribution by $\mathbf{z}$ and the random variable representing the target distribution by $\mathbf{x}$. Further, for notational convenience, we denote the probability density functions (PDFs) of $\mathbf{x}$ and $\mathbf{z}$ by $p$ and $q$, whereas marginal PDFs are denoted by $p_j$ and $q_j$, respectively. Finally, we assume that both random variables $\mathbf{x}$ and $\mathbf{z}$ have continuous and positive density on $\mathbb{R}^D$, i.e $p(x),\, q(z) > 0$ for all $x, z \in \mathbb{R}^D$, where $D$ is the dimensionality of $\mathbf{x}$ and $\mathbf{z}$.

## 2 BACKGROUND

In this section, we give a brief introduction to heavy-tailed distributions and present needed background knowledge about normalizing flows.

### 2.1 HEAVY-TAILED DISTRIBUTIONS

Heavy-tailed distributions are distributions that have heavier tails (i.e. decay slower) than the exponential distribution. Loosely speaking, slowly decaying tails allow to model distributions that generate samples, which differ by a large magnitude from the rest of the samples. For a univariate random variable $\mathbf{x}$ we define heavy-tailedness via its moment-generating function[1]:

**Definition 1** (Heavy-Tailed Random Variables). *Consider a random variable $\mathbf{x} \in \mathbb{R}$ with PDF $p$. Then, we say that $\mathbf{x}$ is heavy-tailed if and only if*

$$\forall \lambda > 0 \; : \; \mathbb{E}_{\mathbf{x}}\big[e^{\lambda \mathbf{x}}\big] = \infty \; .$$

*The function $m_p(\lambda) := \mathbb{E}_{\mathbf{x}}[\exp(\lambda \mathbf{x})]$ is known as the moment-generating function of $\mathbf{x}$. Random variables that are not heavy-tailed are said to be light-tailed.*

Note that this definition is, strictly speaking, merely a definition for heavy right tails. We say a random variable $\mathbf{x} \in \mathbb{R}$ has heavy left tails if $-\mathbf{x}$ has heavy tails according to Definition 1. For simplicity of derivations and w.l.o.g., we proceed with this definition but the derived results can analogously be applied to left tails.

We can assess the degree of tailedness of a distribution. While there are many equivalent notions of the so called tail index, the most straight-forward definition is via the existence of moments:

**Definition 2** (Tail Index). *A random variable $\mathbf{x} \in \mathbb{R}$ with PDF $p$ is said to have tail index[2] $\alpha$ if it holds that*

$$\mathbb{E}_{\mathbf{x}}[|\mathbf{x}|^{\beta}] \begin{cases} < \infty \; , & \textit{if } \beta < \alpha \; , \\ = \infty \; , & \textit{if } \beta > \alpha \; . \end{cases}$$

Since the tail index is tightly related to the decay rate of the PDF, it enables us to assess the degree of heavy-tailedness of a random variable. Therefore, estimation of the tail index became an important objective in extreme value theory and statistical risk assessment (see e.g. Embrechts et al., 2013). Since the existence of the moment does not depend on the "body" of $\mathbf{x}$ but only on the tails of $\mathbf{x}$ (see Proposition 3 in Section A.1 in the Appendix), estimating the tail index by fitting a full parametric model to all data e.g. via likelihood maximization leads to a biased estimator. Instead, semi-parametric estimators have been developed, which aim to fit a distribution only on the tails. Popular methods for tail estimation include the Hill estimator (Hill, 1975), the moment estimator (Dekkers et al., 1989), and kernel-based estimators (Csorgo et al., 1985). In Section B.1 of the Appendix, we discuss these tail estimators and review some practical issues with these.

An example of a heavy tailed distribution is the standardized $t$-distribution, which has parameter $\nu > 0$ referred to as the degree of freedom and a density function given by

$$p(x) := \frac{\Gamma\big(\frac{\nu+1}{2}\big)}{\sqrt{\nu\pi}\Gamma\big(\frac{\nu}{2}\big)}\big(1 + \frac{x^2}{\nu}\big)^{-\frac{\nu+1}{2}} \; , \;\; x \in \mathbb{R} \; ,$$

where $\Gamma$ is the Gamma function. It is known that the $t$-distribution has tail index $\nu$ (see e.g. Kirkby et al. (2019) for a detailed reference).

In the multivariate setting, there exist various definitions of heavy-tailedness. For instance Resnick (2004) make use of a definition based on multivariate regular variation. Jaini et al. (2020) define a multivariate random variable $\mathbf{x}$ to be heavy-tailed if the $\ell_2$-norm is heavy-tailed, a property which we refer to as $\ell_2$-heavy tailed, and which is formally defined as follows:

---

[1]One can readily show that this definition is equivalent to the definition, which compares the tails of $\mathbf{x}$ to the tails of an exponential distribution. See Section 1 in Nair et al. (2013).

[2]Notice that the notion of a tail index is only valid for regularly-varying random variables, which are a subclass of heavy-tailed random variables. For the purpose of this work, it is sufficient to consider regularly varying random variables. More details can be found in Nair et al. (2013).

**Definition 3** ($\ell_2$-Heavy-Tailed). *Let $\mathbf{x} \in \mathbb{R}^D$ be a multivariate random variable. Then, we call $\mathbf{x}$ $\ell_2$-heavy-tailed if it holds that $\|\mathbf{x}\|$ is univariately heavy-tailed according to Definition 1, where $\|\cdot\|$ denotes the $\ell_2$-norm. Otherwise, we call $\mathbf{x}$ $\ell_2$-light-tailed.*

## 2.2 NORMALIZING FLOWS

The fundamental idea behind NFs is based on the change-of-variables formula for probability density functions (PDFs) given in the following theorem.

**Theorem 1** (Change-of-Variables). *Consider random variables $\mathbf{x}, \mathbf{z} \in \mathbb{R}^D$ and a diffeomorphic map $T : \mathbb{R}^D \to \mathbb{R}^D$ such that $\mathbf{x} = T(\mathbf{z})$. Then, it holds that the PDF of $\mathbf{x}$ satisfies*

$$p(x) = q\big(T^{-1}(x)\big)\big|\det J_{T^{-1}}(x)\big| \quad \forall x \in \mathbb{R}^D \ , \tag{1}$$

*where $J_{T^{-1}}(x)$ is the Jacobian of $T^{-1}$ evaluated at $x \in \mathbb{R}^D$.*

This formula allows us to evaluate the possibly intractable PDF of $\mathbf{x}$ if we can evaluate both, the PDF of $\mathbf{z}$ and $T^{-1}(x)$, and efficiently calculate the Jacobian-determinant $\det J_{T^{-1}}(x)$. As $T$ maps $\mathbf{z}$ to $\mathbf{x}$, we denote the distribution of $\mathbf{z}$ and $\mathbf{x}$ as the *base* and the *target distribution*, respectively.

To model the PDF of $\mathbf{x}$ using NFs, it is common to set the base distribution to a standard normal distribution (i.e., $\mathbf{z} \sim \mathcal{N}(0, I)$) and to employ likelihood maximization to learn a parameterized transformation

$$T_\theta := T_\theta^{(L)} \circ \cdots \circ T_\theta^{(1)},$$

which, yet, remains tractable and diffeomorphic. Masked autoregressive flows (MAFs (Papamakarios et al., 2017)) are one popular architecture, which employ transformations $T = (T_1, \ldots, T_D)^\top$ of the form

$$T_j(z) := \mu_j(z_{<j}) + \exp(\sigma_j(z_{<j}))z_j \quad \text{for } j \in \{1, \ldots, D\} \ , \tag{2}$$

where $\mu_j$ and $\sigma_j$ are neural networks, which obtain the first $j-1$ components of $z$ as input and output a scalar. Composing several transformations of the form (2), we obtain the MAF. The autoregressive form in (2) allows us to efficiently evaluate the Jacobian-Determinant due to the diagonal form of $J_T(x)$. A crucial issue of such autoregressive models is that a component $\mathbf{x}_j$ only depends on the previous outputs $\mathbf{x}_{<j}$ and, therefore, they cannot model a causal relationship in which $\mathbf{x}_j$ causes $\mathbf{x}_i$ if $i < j$. This issue can be solved by applying a permutation before each transformation $T_\theta^{(1)}, \ldots, T_\theta^{(L)}$. This is usually a random permutation or the one that reverses the ordering of the components. Therefore, in summary a MAF consists of multiple consecutive layers $T_\theta^{(l)} \circ P^{(l)}$, where $P^{(l)} \in \mathbb{R}^{D \times D}$ is a permutation. MAFs belong to the class of triangular flows, which are defined as flows that consist of diffeomorphisms whose $j$th output only depends on $\mathbf{z}_{\leq j}$. Other examples for triangular flows include RealNVP (Dinh et al., 2017), NAF (Huang et al., 2018), and SOS (Jaini et al., 2019). If the triangular maps are affine linear (such as in (2)), we call the resulting flow a triangular affine flow. Further types of NFs include invertible ResNets (Jacobsen et al., 2018; Behrmann et al., 2019; Chen et al., 2019), continuous flows (Chen et al., 2018; Grathwohl et al., 2019), and many more (Kobyzev et al., 2020).

**Tail-Adaptive Flows.** Jaini et al. (2020) investigated the ability of triangular flows to learn heavy-tailed distributions. The authors have shown that if a triangular affine flow transforms a $\ell_2$-light-tailed distribution, such as the multivariate Gaussian distribution, to a $\ell_2$-heavy-tailed target distribution, then $T_\theta$ cannot be Lipschitz continuous. And more explicitly, it holds the following.

**Theorem 2.** *(Jaini et al., 2020) Let $\mathbf{z}$ be a $\ell_2$-light-tailed random variable and $T$ be an affine triangular flow such that $T_j(z_{\leq j}) = \mu_j(z_{<j}) + \sigma_j(z_{<j})z_j$ for all $j$. If $\sigma_j$ is bounded above and $\mu_j$ is Lipschitz for all $j$, then the transformed variable $\mathbf{x}$ is also $\ell_2$-light-tailed.*

Furthermore, the authors prove that any triangular mapping from an elliptical distribution to a heavier-tailed elliptical distribution must have an unbounded Jacobian-determinant. Clearly, these results illuminate that learning a heavy-tailed distribution using NFs leads to non-Lipschitz transformations and unbounded Jacobians, which inevitably affects training robustness (Behrmann et al., 2021). Motivated by these result, Jaini et al. (2020) propose *Tail-Adaptive Flows* (TAF), which replace the Gaussian base distribution by a multivariate $t$-distribution with one learnable degree of freedom.

## 3 LEARNING THE CORRECT MARGINAL TAIL BEHAVIOR WITH mTAF

In this section, we present a simple extension to triangular affine flows that allows to model distributions with a flexible tail behavior. We start by presenting our theoretical results in Section 3.1. Motivated by these results, we propose *marginally Tail-Adaptive Flow* (mTAF) in Section 3.2.

### 3.1 THE NECESSITY OF A FLEXIBLE BASE DISTRIBUTION

In this work, we investigate the tailedness of NFs more thoroughly through the lense of marginal tailedness, i.e. we consider the univariate tailedness of the marginal distributions of $\mathbf{x}_j$. Therefore, we introduce the following definitions:

**Definition 4** ($j$-Heavy-Tailed, Mixed-Tailed, Fully Heavy-Tailed, Equal Tail Behavior). *We call a random variable $\mathbf{x} \in \mathbb{R}^D$ $j$-heavy-tailed if its $j$th marginal $\mathbf{x}_j$ is heavy-tailed according to Definition 1. Otherwise, we call $\mathbf{x}$ $j$-light-tailed. $\mathbf{x}$ is said to be mixed-tailed if there exists $j_1, j_2$ such that $\mathbf{x}$ is $j_1$-heavy-tailed and $j_2$-light-tailed. Further, we say that $\mathbf{x}$ is fully heavy-tailed if $\mathbf{x}$ is $j$-heavy-tailed for all $j \in \{1, \dots, D\}$. We define two random variables $\mathbf{x}$ and $\mathbf{z}$ to have equal tail behavior if it holds for all $j$ that*

$$\mathbf{x} \text{ is } j\text{-heavy tailed} \quad \Leftrightarrow \quad \mathbf{z} \text{ is } j\text{-heavy tailed} \ .$$

We found the following relation to Definition 3.

**Proposition 1** ($j$-Heavy-Tailedness induces $\ell_2$-Heavy-Tailedness). *Assume that $\mathbf{x}$ is $j$-heavy-tailed for any $j$. Then, $\mathbf{x}$ is also $\ell_2$-heavy-tailed.*

The proof can be found in Section A.1 in the Appendix. The proposition shows that $j$-heavy-tailedness is a more general notion of multivariate heavy-tailedness than $\ell_2$-heavy-tailedness, which allows a narrow inspection of the tail behavior. More precisely, the new notion allows us to differentiate between fully heavy-tailed random variables and mixed-tailed random variables, which are both $\ell_2$-heavy-tailed. One can now wonder how the tails, described by the novel notation, behave for NFs, which we answer by the following two results. The first result states that, under mild technical conditions, fully heavy-tailedness of the base distribution is preserved by triangular affine maps.

**Proposition 2** (Triangular Affine Maps preserve Fully Heavy-Tailedness). *Let $\mathbf{z}$ be a fully heavy-tailed random variable that satisfies Assumption 1[3] and let $T$ be a a triangular affine map, that is, $T_j(z_j, z_{<j}) = \mu_j(z_{<j}) + \sigma_j(z_{<j})z_j$ with $\sigma_j > 0$. Then, it holds that $T(\mathbf{z})$ is also fully heavy-tailed.*

A formal proof can be found in Section A.2 of the Appendix. Assumption 1 is a mild condition on the decay rate of the copula density of $\mathbf{z}$. We explain this condition in more detail and give various examples in Section A.3 of the Appendix.

It is clear that permuting the marginals does not change the heavy-tailedness. Hence, by iterative application of Proposition 2, we deduce that affine triangular flows with a fully heavy-tailed base distribution are unable to model mixed tailed distributions. Implicitly, Proposition 2 states that a Lipschitz normalizing flow as proposed by Jaini et al. (2020) is not able to model mixed-tailed distributions. However, the following Theorem that we consider as our main result guides us towards a flow architecture that is able to model target distributions that are mixed-tailed in a marginally tail-adaptive way.

**Theorem 3** (Learning the correct Tail Behavior). *Consider a random-variable $\mathbf{z}$ that is $j$-light-tailed for $j \in \{1, \dots, d\}$ for some $d < D$ and $j$-heavy-tailed for $j \in \{d+1, \dots, D\}$. Then, under the same conditions as in Theorem 2 and Proposition 2, it holds that $\mathbf{z}$ and $T(\mathbf{z})$ have the same tail behavior.*

*Proof.* Since the result combines Theorem 2 and Proposition 2 in an evident fashion, we just quickly present a sketch of the proof. First, let us consider $j \leq d$. Then it holds for the moment-generating function of $\mathbf{x}_j$ that

$$m_{\mathbf{x}_j}(\lambda) = \int_{\mathbb{R}^D} e^{\lambda T_j(z_{\leq j})} q(z) dz = \int_{\mathbb{R}^j} e^{\lambda T_j(z_{\leq j})} p_{\leq j}(z_{\leq j}) dz_{\leq j} \ ,$$

---

[3]This Assumption can be found in Section A.2 in the Appendix.

which has been shown to be bounded for some $\lambda > 0$ (see the proof of Theorem 2 in Jaini et al. (2020)). Therefore, $\mathbf{x}$ is $j$-light-tailed for all $j \leq d$. In the case $j > d$, we notice[4] that the proof for heavy-tailedness of $T_j(\mathbf{z}_{\leq j})$ involves just the heavy-tailedness of $\mathbf{z}_j$ and not of any other component of $\mathbf{z}_{<j}$. Hence, if $\mathbf{z}_j$ is heavy-tailed, then $\mathbf{x}_j = T_j(\mathbf{z}_{\leq j})$ is also heavy-tailed, regardless of $\mathbf{z}_{<j}$. Therefore, $\mathbf{x}$ is $j$-heavy-tailed for all $j > d$, which completes the proof. Note that in general we cannot deduce the latter conclusion for light-tailed marginals, i.e. if $\mathbf{z}_j$ is light-tailed, this does not mean that $\mathbf{x}_j$ is also light-tailed. This is only the case, if all $\mathbf{z}_{<j}$ are light-tailed as well. □

### 3.2 MARGINALLY TAIL-ADAPTIVE FLOW (MTAF)

Our main result, Theorem 3, prompts that if we maintain an ordering of the marginals such that the first marginals are light-tailed and the following are heavy-tailed in each flow step, we retain the marginal tail behavior of the base distribution in the estimated target distribution. This finding motivates the novel NF proposed in this paper. The proposed approach combines research findings from extreme value theory (Embrechts et al., 2013; Nair et al., 2013), recent findings about normalizing flows (Jaini et al., 2020; Alexanderson & Henter, 2020; Laszkiewicz et al., 2021), and the results presented herein. The proposed mTAFs consists out of three steps depicted in Figure 1 and described in the following:

**Step 1: Estimating the marginal tail indices and defining the marginal distributions.** For each marginal, i.e. for the marginal distribution $q_j$ of each $\mathbf{x}_j$, $j = 1, \ldots, D$, we use the moments double-bootstrap estimator (Draisma et al., 1999) and the kernel-type double-bootstrap estimator (Groeneboom et al., 2003) to assess heavy-tailedness of the data distribution. If both estimators predict a light-tailed distribution, we set the corresponding marginal base distribution $q_j$ to be standard normal distributed, i.e. $\mathbf{z}_j \sim \mathcal{N}(0, 1)$. Otherwise we set the marginal to the standardized $t$-distribution with the estimated degree of freedom, i.e. $\mathbf{z}_j \sim t_{\hat{\nu}_j}$, where $\hat{\nu}_j$ is the Hill double-bootstrap estimator (Danielsson et al., 2001; Qi, 2008). In Section B.1 of the Appendix, we present more details about the tail-assessment scheme.

**Step 2: Defining the base distribution.** We construct the base distribution as the mean-field approximation of the marginals, i.e. $\mathbf{z}$ has the density $q(z) := \prod_{j=1}^{D} q_j(z_j)$ with marginal densities $q_j$ defined in step 1. Further, to satisfy the assumptions of Theorem 3, we need to permute the marginals such that it holds $\mathbf{z}_j \sim \mathcal{N}(0, 1)$ for $j \leq d$ and $\mathbf{z}_j \sim t_{\hat{\nu}_j}$ for $j > d$. We apply the same permutation to restructure our data according to the base components. To account for tail index estimation errors and for more flexible learning, we make the tail indices (i.e. the degrees of freedom of each $t$-distribution) learnable. That is, we initialize the degree of freedom of the $j$th marginal with $\hat{\nu}_j$ but adapt the parameter together with the network parameters throughout training.

**Step 3: A data-driven permutation scheme.** Recall, that vanilla triangular flows employ a permutation step after each transformation to enhance the mixing of variables. However, purely random permutations might lead to a violation on the ordering of marginals, which is necessary to ensure Theorem 3. Therefore, we permute only within the set of heavy-tailed marginals and within the set of light-tailed marginals, to ensure the validity of Theorem 3. Within these groups one can choose any permutation scheme.

Without loss of generality, we assume that the first $d$ components of $\mathbf{z}$ are light-tailed and the remaining $D - d$ components are heavy-tailed[5]. Then, the training objective is to optimize for flow parameters $\hat{\theta}$ and degrees of freedom $\hat{\nu} = [\hat{\nu}_{d+1}, \ldots, \hat{\nu}_D]$ to maximize the log-likelihood

$$L(\hat{\theta}, \hat{\nu}; X) = \sum_{j=1}^{N} \left\{ \log\left( \prod_{i=1}^{d} \pi\left(T_{\hat{\theta}}^{-1}\left(x^{(j)}\right)_i\right) \cdot \prod_{i=d+1}^{D} t_{\hat{\nu}_i}\left(T_{\hat{\theta}}^{-1}\left(x^{(j)}\right)_i\right) \right) - \log \det J_{T_{\hat{\theta}}}\left(x^{(j)}\right) \right\}$$

$$= \sum_{j=1}^{N} \left\{ \sum_{i=1}^{d} \log \pi\left(T_{\hat{\theta}}^{-1}\left(x^{(j)}\right)_i\right) + \sum_{i=d+1}^{D} \log t_{\hat{\nu}_i}\left(T_{\hat{\theta}}^{-1}\left(x^{(j)}\right)_i\right) - \log \det J_{T_{\hat{\theta}}}\left(x^{(j)}\right) \right\}$$

where $X := (x^{(1)}, \ldots x^{(N)})$ is the data, and $\pi$ and $t_{\hat{\nu}}$ are the PDF of the standard normal distribution and the standard $t$-distribution with $\hat{\nu}$ degrees of freedom, respectively.

---

[4]For details, we refer to the proof of Proposition 2 in the Appendix.
[5]Otherwise we permute the marginals as described in Step 2.

When applying our theoretical results presented in the previous section to the proposed mTAF, we can show that it fulfills the desired tail-preserving property, as formalized by the following corollary:

**Corollary 1** (Marginal Tail-Adaptive). *Under the same assumptions as in Theorem 2 and in Proposition 2, mTAFs are marginally tail-adaptive, that is, $\mathbf{z}$ and $\mathbf{x} = T(\mathbf{z})$ have the same tail behavior.*

## 4    EXPERIMENTAL ANALYSIS ON SYNTHETIC DATA

To investigate the benefits of mTAF we perform an empirical analysis in which we conduct experiments on synthetic heavy-tailed data. We construct synthetic 16-dimensional data sets and investigate eight different settings: The marginals are chosen to contain $h \in \{1, 2, 4, 8\}$ mixtures of two $t$-distributions, four Gaussian distributions, two mixtures of three Gaussian distributions, and $10 - h$ mixtures of two Gaussian distributions. The degree of freedom of all $t$-distributions is either $\nu = 2$ or $\nu = 3$. In all mixtures all mixture components are weighted equally, means are uniformly sampled from $[4, 4]$, and standard-deviations are sampled from $[1, 2]$. Using a Gaussian Copula, we construct a complex joint distribution with mixed-tailed marginals. A detailed description of the data set generation can be found in Section B.2 in the Appendix. Details about hyperparameters can be found in Section B.4 in the Appendix. We provide a PyTorch implementation and the code for all experiments along the submission.

### 4.1    MODEL ACCURACY

In our first experiment, we compare the performance of 5 different MAFs: a vanilla flow with $\mathbf{z} \sim \mathcal{N}(0, I)$, the TAF, a variation of TAF (TAF($\hat{\nu}_1, \dots, \hat{\nu}_D$) in which all marginals have their own independent degree of freedom (i.e. $\mathbf{z}_j \sim t_{\hat{\nu}_j}(0, 1)$ with trainable degrees of freedom $\hat{\nu}_j$), and mTAF as described in Section 3.2 with learnable and fixed tail indices, where the letter is denoted as mTAF (fixed $\hat{\nu}$).

Table 1 summarizes the test losses archived by the five models in the eight different settings. The results demonstrate that mTAF can efficiently learn a multivariate heavy-tailed distribution, whereas the vanilla approach, i.e. fixing $\mathbf{z} \sim \mathcal{N}(0, I)$, is prone to optimization instabilities (as can be seen by high standard deviations) and leads to higher values of the negative log-likelihood. Moreover, TAF performs worse than mTAF (and is also less stable) which could be contributed to having only one joint degree of freedom to model the tailedness of all marginals. Using independent degrees of freedom however only leads to slight improvements, illustrating the necessity of the proposed permutation scheme of mTAF. The performance increase is especially severe for marginals with small tails index, i.e., when the degree of freedom of the marginals of the data-generating distribution is $\nu = 2$. Fixing the tail indices to the estimated values already leads to great results, which can further be increased by adapting the parameters during training.

### 4.2    GENERATING TAIL EVENTS

An often-used property of NFs is the diffeomorphic relation between base and target distribution, which can be used, for instance, for image interpolation[6] or for constructing counterfactuals[7]. In this work, we present another practical advantage, which we motivate by the derived theory, namely the generation of tail samples. One generic way to generate tail samples in the target space is to generate tail samples in the base space, which are then pushed towards the target space. However, in this work we propose to solve a more specific problem, which is the generation of samples that are from the $j$-tails, i.e. samples from the joint distribution of $\mathbf{x}$ with the $j$th component $\mathbf{x}_j$ being from the tail of the $j$th marginal. Recall that in the proof of Proposition 2 we show that if $\mathbf{z}_j$ is heavy-tailed, then $T_j(\mathbf{z}_{\leq j})$ resulting from applying one affine triangular flow layer is also heavy-tailed, independently of $\mathbf{z}_{<j}$. Inspired by this result, we hypothesize that we can generate multivariate samples $x$ with heavy $j$-tails by sampling tail events of $\mathbf{z}_j$, complementing them with remaining components $z_{\neq j}$ sampled from $\mathbf{z}_{\neq j}$, and pushing the resulting $z$ through the flow, which retains the $j$-heavy-tailedness. However, since we apply a permutation after each layer, to sample from the

---

[6]by interpolating in the base distribution and pushing them forward using the NF (Kingma & Dhariwal, 2018).

[7]by applying gradient ascent in the base space (Dombrowski et al., 2021).

Table 1: Average test loss (lower is better) on the synthetic data in various settings over 10 trials. The error bars represent one standard deviation and the numbers in brackets denote the number of crashed runs. Models that are significantly better than the other models (according to a one-sided $t$-test) are highlighted.

| | number of heavy-tailed components $h$ | | | |
|---|---|---|---|---|
| | 1 | 2 | 4 | 8 |
| **$\nu = 2$** | | | | |
| Vanilla | $35.68 \pm 1.92$ | $36.68 \pm 1.77$ | $38.19 \pm 1.71$ | $45.78 \pm 13.91$ |
| TAF | $36.29 \pm 0.61$ (1) | $35.54 \pm 1.08$ (2) | $36.04 \pm 0.36$ | $38.33 \pm 0.40$ |
| TAF$(\hat{\nu}_1, \ldots, \hat{\nu}_D)$ | $35.23 \pm 0.76$ | $35.69 \pm 0.44$ (1) | $36.44 \pm 0.58$ | $38.63 \pm 0.36$ |
| mTAF (fixed $\hat{\nu}$) | $\mathbf{33.48 \pm 0.04}$ | $34.17 \pm 0.07$ | $\mathbf{35.46 \pm 0.09}$ | $38.52 \pm 0.36$ |
| mTAF | $\mathbf{33.48 \pm 0.04}$ | $\mathbf{34.11 \pm 0.06}$ | $\mathbf{35.39 \pm 0.13}$ | $38.44 \pm 0.28$ |
| **$\nu = 3$** | | | | |
| Vanilla | $33.47 \pm 0.20$ | $33.80 \pm 0.19$ | $34.43 \pm 0.22$ | $36.09 \pm 0.13$ |
| TAF | $33.81 \pm 0.37$ | $34.16 \pm 0.45$ | $34.48 \pm 0.20$ | $35.75 \pm 0.09$ |
| TAF$(\hat{\nu}_1, \ldots, \hat{\nu}_D)$ | $33.54 \pm 0.17$ | $33.83 \pm 0.19$ | $34.63 \pm 0.58$ | $35.80 \pm 0.15$ |
| mTAF (fixed $\hat{\nu}$) | $\mathbf{33.34 \pm 0.04}$ | $\mathbf{33.66 \pm 0.04}$ | $34.28 \pm 0.07$ | $35.80 \pm 0.09$ |
| mTAF | $\mathbf{33.31 \pm 0.05}$ | $\mathbf{33.63 \pm 0.04}$ | $\mathbf{34.21 \pm 0.04}$ | $\mathbf{35.63 \pm 0.08}$ |

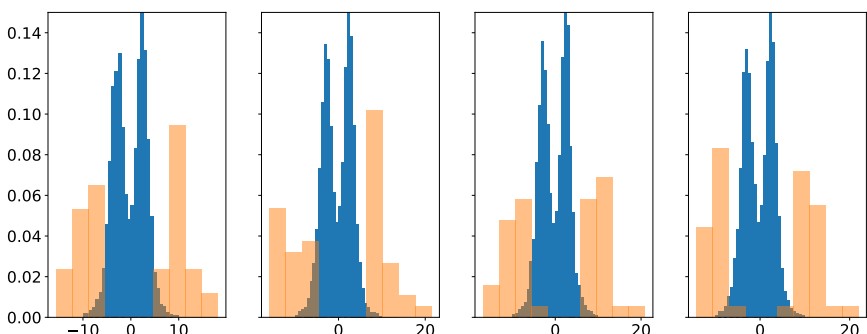

Figure 2: Visualizing the 16th marginal of samples resulting from the tail generation procedure from Section 4.2 in orange in the settings $h \in \{1, 2, 4, 8\}$ and $\nu = 2$ (from left to right). In blue, we visualize samples from the true distribution.

$j$th marginal of the target distribution, we need to employ the above procedure with tail samples of $\mathbf{z}_{k^{-1}(j)}$, where $k : \{1, \ldots, D\} \to \{1, \ldots, D\}$ is the permutation mapping resulting from applying all permutation layers, and $k^{-1}$ is its inverse.

For all models trained on the data sets with $\nu = 2$, we generate $1\,000$ samples of the heavy-tailed variable $\mathbf{z}_{k^{-1}(16)}$ and consider the 50 with largest absolute value as tail samples. In Figure 2 we visualize the results for mTAF: We depict the 16th component $\mathbf{x}_{16}$ of the samples generated by the described procedure for the distributions with different amounts of heavy-tailed components. It becomes clear that the sampling procedure indeed results only in samples from the joint distribution where the 16th component stems from the tails of the corresponding marginal. Surprisingly, even in the case in which 8 heavy-tailed components are permuted (i.e. $h = 8$) the sampling procedure succeeds by ignoring the body of the distribution and generating exclusively tail samples in the 16th component. As a further proof of concept, we demonstrate in Figure 5 in Section B.3 in the Appendix that sampling tail events from other marginals than $\mathbf{z}_{k^{-1}(16)}$ does not result in tail events in the 16th component of $\mathbf{x}$. Hence, this result substantiates our intuition that $\mathbf{z}_{k^{-1}(j)}$ is the "responsible" factor for generating $j$-tail samples. Furthermore, we rerun the sampling procedure with a vanilla flow and a TAF, which we visualize in Figure 3. While TAF is able to generate

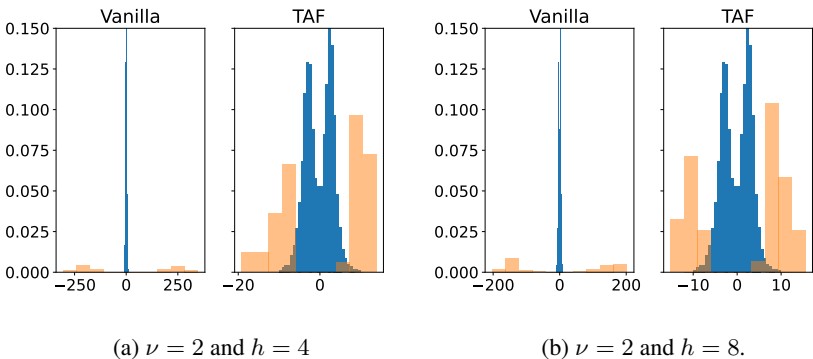

(a) $\nu = 2$ and $h = 4$                    (b) $\nu = 2$ and $h = 8$.

Figure 3: Applying the tail sample generation procedure form Section 4.2 with a vanilla flow and a TAF in two different settings. Samples from the true distribution are visualized in blue and flow samples are in orange.

reasonable tail samples comparable to mTAF, the vanilla flow is generating extreme outliers, and is thus, not able to efficiently sample from the tails.

## 5 CONCLUSION AND FUTURE RESEARCH DIRECTIONS

In this work, we deepen the mathematical understanding of the tail behavior of triangular flows. We note that the distribution we want model may have heavy- as well as light-tailed marginals, and show how the marginal tail behavior of the target distribution of the flow relates to the tail behavior of its base distribution. Based on these theoretical findings we propose a new algorithm, which we refer to as mTAF. In particular, we initialize a trainable base distribution based on statistical tail estimates of the target, and employ a data-driven permutation scheme that guarantees the correct tail behavior of the target distribution. An in-depth empirical analysis on synthetic data with heavy- and light-tailed marginals shows that mTAF benefits in terms of estimation performance and robustness of the training. Moreover, we provide a sampling strategy inspired by out theory that allows to generate joint samples from the target distribution with a customizable tail behavior. In summary, we introduce a novel way to impose an inductive bias in normalizing flows that allows to model the tails of the target distribution.

Yet, we think that there exists much potential for interesting follow-up works. For instance, we are aware that most SOTA architectures are indeed non-affine, such as NSF (Durkan et al., 2019), which are based on monotone rational-quadratic splines, or SOS (Jaini et al., 2019), which are based on strictly increasing polynomials. However, similarly to (Jaini et al., 2020), we hypothesize that the trade-off between a flexible base distributions and the complexity of the flow leads to simpler and more efficient models. This, and the investigation on the impact of mTAF on real world data remains subject of further future studies. Furthermore, the attentive might have noticed that the proposed mTAF employs a restricted set of permutations, which limits its generality. One possible solution for this issue is to employ more expressive base distributions that introduce marginal dependencies, which extends the generality of mTAF, while preserving its ability to be marginally tail-adaptive. For modeling these more expressive base distributions it is evident to study copula theory, which allows a grounded treatment of various kinds of multivariate dependency concepts, even beyond the scope of pearson correlation, such as tail dependencies (Joe, 2014). In addition, we believe that mTAFs, and further improvements of them, bear lots of potential for sampling methods, which benefit from the whole range of samples from the target distribution, including tail samples. Generation of such samples might improve methods for simulation-based inference (Cranmer et al., 2020) such as VAEs and are essential for the analysis of extremes such as in weather forecasting.

Last but not least, we want to highlight that our theory is not limited to maximum likelihood training and one could think of various other training paradigms, for instance based on $f$-divergences. Using alternative optimization metrics, we can potentially put more emphasize on the learning of the correct tail behavior.

**Reproducibility Statement**   This work concentrates on the mathematical understanding of normalizing flows, and in doing so, includes technical proofs, which might rely on abstract assumptions. Nonetheless, we put our best effort in making all the proofs and the derived theory in Section A.1 and A.2 precise, accessible, and correct. To avoid misconceptions, we dedicate Section A.3 to the explanation and clarification of Assumption 1. We provide documented source code including code execution instructions, which allows the reproducibility of our empirical results. Additionally, all algorithms and hyperparameters are presented in Section B.

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

## A  THEORY AND PROOFS

In this Section, we derive and proof all of our theoretical results. We start to present some preliminary results in Section A.1, which help us providing the proof of our main result in Section A.2. In Section A.3, we illuminate the technical Assumption 1 and provide examples and intuitions.

### A.1  PRELIMINARY THEORETICAL RESULTS

**Proposition 3.** *Let $\mathbf{x} \in \mathbb{R}$ be a random variable. Then, it holds that $\mathbf{x}$ has tail index less or equal than $\alpha$ iff. for any $\beta > \alpha$ and any $C > 0$ it is*

$$\int_{|x|>C} |x|^\beta p(x)dx = \infty \ .$$

*Proof.* Let us first assume that $\mathbf{x}$ has tail index at most $\alpha$. Then, according to Definition 2, we know that for any $\beta > \alpha$ and $C > 0$

$$\infty = \mathbb{E}_{\mathbf{x}}[|\mathbf{x}|^\beta] = \int_{\mathbb{R}} |x|^\beta p(x)dx = \int_{|x|\leq C} |x|^\beta p(x)dx + \int_{|x|>C} |x|^\beta p(x)dx$$

$$\leq C^\beta \int_{|x|\leq C} p(x)dx + \int_{|x|>C} |x|^\beta p(x)dx \ .$$

Since we assume $p$ to be continuous, we can bound $p$ on the compact interval $[-C, C]$, and hence, the first above integral must be bounded. Therefore, it is

$$\int_{|x|>C} |x|^\beta p(x) dx = \infty .$$

To prove the back-direction, let us consider a $\beta > \alpha$ and $C > 0$. Very similarly to the forward-proof, we can now see that

$$\begin{aligned}
\infty &= \int_{|x|>C} |x|^\beta p(x) dx \\
&= C^\beta \int_{|x|\leq C} p(x) dx + \int_{|x|>C} |x|^\beta p(x) dx \\
&= \mathbb{E}_{\mathbf{x}}\big[|\mathbf{x}|^\beta\big] ,
\end{aligned}$$

where the second equality follows from the finiteness of the integral. Since this follows for any $\beta > \alpha$, $\mathbf{x}$ must have tail index $\alpha$ or less.

$\square$

This simple result demonstrates that the tail index, as indicated by the name, depends on the tail of the distribution, i.e. on the PDFs behavior for large values $|x| > C$. This fact motivates why maximum likelihood estimations of the tail index, which depend on the whole distribution are biased. For more elaborate details, we refer to Section 9 in Nair et al. (2013).

In a similar fashion to the previous result, the next technical lemma states that unboundedness of the moment-generating function is due to the unboundedness of the integrand for tail events, i.e. for $z > z^*$. This little lemma turns out to be useful in the proof of Proposition 2.

**Lemma 1.** *Let $\mathbf{z} \in \mathbb{R}$ be heavy-tailed. Then it holds for any $z^* \in \mathbb{R}$ and $\lambda > 0$ that*

$$\int_{z>z^*} e^{\lambda z} p_{\mathbf{z}}(z) dz = \infty .$$

*Proof.* Since $\mathbf{z} \in \mathbb{R}$ is heavy-tailed, we know that for all $\lambda > 0$

$$\begin{aligned}
\infty = m_{\mathbf{z}}(\lambda) &= \int_{\mathbb{R}} e^{\lambda z} p_{\mathbf{z}}(z) dz \\
&= \int_{z\leq z^*} e^{\lambda z} p_{\mathbf{z}}(z) dz + \int_{z>z^*} e^{\lambda z} p_{\mathbf{z}}(z) dz \\
&\leq F_{\mathbf{z}}(z^*) e^{\lambda z^*} + \int_{z>z^*} e^{\lambda z} p_{\mathbf{z}}(z) dz ,
\end{aligned}$$

where $F_{\mathbf{z}}$ is the CDF of $\mathbf{z}$. The last inequality follows from the fact that $\exp(\lambda z)$ is monotonic increasing in $z$. Since the first summand is bounded, it follows that the second summand must be unbounded. This completes the proof. $\square$

Recall that in Proposition 1 we state that $j$-heavy-tailedness induces $\ell_2$-heavy-tailedness. In the following, we provide a formal proof of this result.

*Proof of Proposition 1.* For this proof, we employ the equivalent definition of heavy-tailedness of $\mathbf{x}_j$ via the decay rate of its distribution function (see e.g. Lemma 1.1. in Nair et al. (2013)), i.e.

$$\limsup_{x_j \to \infty} \frac{1 - F_j(x_j)}{e^{-\lambda x_j}} = \infty \quad \text{for all } \lambda > 0 , \tag{3}$$

where $F_j$ is the CDF of $\mathbf{x}_j$. Since $x_j \leq \|x\|$ for all $x \in \mathbb{R}^D$, we can conclude that $F_j(a) \geq F_{\|x\|}(a)$ for $a \in \mathbb{R}$. Therefore,

$$\frac{1 - F_{\|\mathbf{x}\|}(a)}{e^{-\lambda a}} \geq \frac{1 - F_{x_j}(a)}{e^{-\lambda a}} \to \infty \quad \text{for } a \to \infty .$$

According to the equivalent definition in (3), $\|\mathbf{x}\|$ is heavy-tailed, which proves that $\mathbf{x}$ is $\ell_2$-heavy-tailed. $\square$

The following is a well-known implication of the change of variables formula and the integration rule by substitution, which we are going to apply in the subsequent proofs.

**Lemma 2** (Substitution in the Moment-Generating Function). *Let $T$ be a diffeomorphism such that $T(\mathbf{z}) = \mathbf{x}$ for some random variables $\mathbf{x}, \mathbf{z} \in \mathbb{R}^D$. Then, we can rewrite*

$$\int_{\mathbb{R}^D} e^{\lambda x} p(x) dx = \int_{\mathbb{R}^D} e^{\lambda T(z)} q(z) dz \ .$$

For completeness, we give a brief proof of this result.

*Proof.* Using the change of variables formula, see (1), we can write

$$\int_{\mathbb{R}^D} e^{\lambda x} p(x) dx = \int_{\mathbb{R}^D} e^{\lambda x} q\big(T^{-1}(x)\big) \big|\det J_{T^{-1}(x)}\big| dx \ .$$

Now, we can rewrite $\exp(\lambda x) = \exp(\lambda T\big(T^{-1}(x)\big)$ and substitute $z = T^{-1}$. Integration by substitution completes the proof. $\square$

Next, we present how we can use copulae to reformulate a multivariate PDF.

**Definition 5** (Copula). *A copula is a multivariate distribution with cumulative distribution function (CDF) $C : [0,1]^D \to [0,1]$ that has standard uniform marginals, i.e. the marginals $C_j$ of $C$ satisfy $C_j \sim U[0,1]$.*

**Theorem 4** (Sklar's Theorem). *Taken from Hofert et al. (2018).*

1. *For any $D$-dimensional CDF $F$ with marginal CDFs $F_1, \ldots, F_D$, there exists a copula $C$ such that*

$$F(z) = C\big(F_1(z_1), \ldots, F_D(z_D)\big) \tag{4}$$

*for all $z \in \mathbb{R}^D$. The copula is uniquely defined on $\mathcal{U} := \prod_{j=1}^{D} \mathrm{Im}(F_j)$, where $\mathrm{Im}(F_j)$ is the image of $F_j$. For all $u \in \mathcal{U}$ it is given by*

$$C(u) = F\big(F_1^{\leftarrow}(u_1), \ldots, F_D^{\leftarrow}(u_D)\big) \ ,$$

*where $F_j^{\leftarrow}$ are the right-inverses of $F_j$.*

2. *Conversely, given any $D$-dimensional copula $C$ and marginal CDFs $F_1, \ldots F_D$, a function $F$ as defined in (4) is a $D$-dimensional CDF with marginals $F_1, \ldots, F_D$.*

Therefore, if $F$ is absolutely continuous, we can differentiate (4) to obtain the PDF of $\mathbf{z}$

$$q(z) = c\big(F_1(z_1), \ldots, F_D(z_D)\big) \prod_{j=1}^{D} q_j(z_j) \ ,$$

where $c$ denotes the PDF of the copula $C$.

Lastly, we present the following asymptotic behavior of the inverse CDF of a standard Gaussian distribution, which we use in Section A.3 to explain Assumption 1.

**Lemma 3** (Asymptotic Behavior[8] of $\Phi^{-1}(1-y)$). *Denote by $\Phi$ the CDF of a standard Gaussian distribution. Then, it holds for the inverse of $\Phi$ that*

$$\Phi^{-1}(1-y) \sim \sqrt{-2\log(y)} \quad \text{for } y \to 0 \ .$$

*Proof.* First, we note that

$$\Phi(x) = \frac{1}{2} + \frac{1}{2} \mathrm{erf}\left(\frac{x}{\sqrt{2}}\right) \sim 1 - \frac{1}{x\sqrt{2}} e^{-x^2/2} \ ,$$

---

[8]The idea of the proof is due to (https://math.stackexchange.com/users/491644/maxim)

which is a well-known asymptotic (Liu et al., 2012). Here, $\mathrm{erf}$ denotes the error-function. Rearranging terms gives

$$\log\big(1 - \Phi(x)\big) \sim -\log\Big(x\sqrt{2\pi}\Big) - \frac{x^2}{2} \sim -\frac{x^2}{2} \quad \text{as } x \to \infty \ .$$

Finally, we can invert the above asymptotic equation to obtain

$$\Phi^{-1}(y) = \sqrt{-2\log(1-y)} \quad \text{for } y \to 1$$

or equivalently

$$\Phi^{-1}(1-y) = \sqrt{-2\log(y)} \quad \text{for } y \to 0 \ .$$

$\square$

### A.2 PROOF OF THE MAIN RESULT

The proof of Proposition 2 relies on lower-bounding the moment-generating function of each marginal $\mathbf{x}_j$. In order to derive such a bound of a multivariate integral, we rewrite the joint distributions $q_{\leq j}$ using their copula densities:

$$q_{\leq j}(z_{\leq j}) = c_j\big(F_1(z_1), \ldots, F_j(z_j)\big) \prod_{i < j} q_i(z_i)$$

for any $j \in \{1, \ldots, D\}$ and for corresponding copula density $c_j$. Our proof relies on the following technical condition on the decay rate of the copula densities.

**Assumption 1** (Bounding the Marginal Decay Rate of the Copula Densities). *For all* $j \in \{1, \ldots, D\}$ *and* $\lambda > 0$ *there exists a compact set* $\mathbb{S} \subset \mathbb{R}^{j-1}$ *with positive (Lebesgue-)mass, a constant* $z_j^* > 0$, *a scaling constant* $s > 0$, *and a function* $f(z_{<j}) < \lambda\sigma(z_{<j})$ *for* $z_{<j} \in \mathbb{S}$ *such that*

$$c_j\big(F_1(z_1), \ldots, F_j(z_j)\big) \geq se^{-f(z_{<j})z_j} \quad \text{for } z_j > z_j^* \text{ and } z_{<j} \in \mathbb{S} \ , \tag{5}$$

*where* $c_j$ *is the copula density of* $q_{\leq j}$

This assumption sets a bound on the decay rate of the copula density with respect to $z_j$. We clarify this assumption in Section A.3 with additional examples.

Now, we set all preliminaries to prove Proposition 2.

*Proof of Proposition 2.* We start by considering the case $j = 1$. In this case it is $\mathbf{x}_1 = \mu + \sigma\mathbf{z}_1$ and therefore

$$m_{\mathbf{x}_1}(\lambda) = \int_{\mathbb{R}} e^{\lambda x_1} p_1(x_1) dx_1$$

$$= \int_{\mathbb{R}} e^{\lambda(\mu_1 + \sigma_1 z_1)} q_1(z_1) dz_1 \quad \text{(Lemma 2)}$$

$$= e^{\lambda\mu_1} \int_{\mathbb{R}} e^{\lambda\sigma_1 z_1} q_1(z_1) dz_1 \ .$$

Defining $\lambda' := \lambda\sigma_1 > 0$, we can see that the last integral is unbounded due to the heavy-tailedness of $\mathbf{z}_1$, see Definition 1. Therefore, $m_{\mathbf{x}_1}(\lambda) = \infty$ for all $\lambda > 0$, which proves the heavy-tailedness of $\mathbf{x}_1$.

Next, we consider the case $j > 1$. Again, we examine the moment-generating function of $\mathbf{x}_j$. Define the $j$th canonical basis vector $v_j := (0, \ldots, 0, 1, 0, \ldots, 0)^\top$. Then,[9]

$$m_{\mathbf{x}_j}(\lambda) = m_{v_j^\top \mathbf{x}} = \int_{\mathbb{R}^D} e^{\lambda v_j^\top x} p(x) dx \quad \text{(LOTUS)}$$

$$= \int_{\mathbb{R}^D} e^{\lambda T_j(z_j, z_{<j})} q(z) dz \quad \text{(Lemma 2)}$$

$$= \int_{\mathbb{R}^j} e^{\lambda\mu(z_{<j}) + \lambda\sigma(z_{<j})z_j} q_{\leq j}(z_{\leq j}) dz_{\leq j}$$

$$= \int_{\mathbb{R}^{j-1}} e^{\lambda\mu(z_{<j})} \int_{\mathbb{R}} e^{\lambda\sigma(z_{<j})z_j} q_{\leq j}(z_{\leq j}) dz_j dz_{<j} \ . \tag{6}$$

---

[9]Note that for the sake of clarity, we leave out the index $j$ in $\mu_j$ and $\sigma_j$.

Using Sklar's Theorem (Theorem 4), we can write any joint PDF as the product of marginals and a copula density $c_j$ such that

$$q_{\leq j}(z_{\leq j}) = c_j\big(F_1(z_1), \ldots, F_j(z_j)\big) \prod_{i<j} q_i(z_i) \ . \tag{7}$$

We plug (7) into (6) to obtain

$$m_{\mathbf{x}_j}(\lambda) = \int_{\mathbb{R}^{j-1}} e^{\lambda\mu(z_{<j})} q_{<j}(z_{<j}) \int_{\mathbb{R}} e^{\lambda\sigma(z_{<j})z_j} c_j\big(F_1(z_1), \ldots, F_j(z_j)\big) q_j(z_j) dz_j dz_{<j}$$

$$\geq \int_{\mathbb{S}} e^{\lambda\mu(z_{<j})} q_{<j}(z_{<j}) \int_{z_j>z_j^*} e^{\lambda\sigma(z_{<j})z_j} c_j\big(F_1(z_1), \ldots, F_j(z_j)\big) q_j(z_j) dz_j dz_{<j} \ , \tag{8}$$

since all quantities within the integral are positive. Using Assumption 1, we can bound the inner integral of the above equation, which we denote by $A(z_{<j})$, and get

$$A(z_{<j}) \geq s \int_{z_j>z_j^*} e^{(\lambda\sigma(z_{<j})-f(z_{<j}))z_j} q_j(z_j) dz_j$$

$$= s \int_{z_j>z_j^*} e^{\lambda'z_j} q_j(z_j) dz_j \quad (\text{define } \lambda' := \lambda - \sigma(z_{<j}) - f(z_{<j}) )$$

$$= \infty \quad \text{for all } z_{<j} \in \mathbb{S} \ ,$$

due to the heavy-tailedness of $\mathbf{z}_j$ and Lemma 1. Since $\mathbb{S}$ is compact, $\mu$ and $q$ are both continuous, and $q$ is positive, we deduce that $\exp(\lambda\mu(z_{<j}))q_{<j}(z_{<j})$ is lower-bounded (by a constant larger than 0) in $\mathbb{S}$. Therefore, employing (8) and using that $\mathbb{S}$ has positive mass, we can lower-bound the moment-generating function by $\infty$, which proves the heavy-tailedness of $\mathbf{x}_j$. In summary, $\mathbf{x}$ is $j$-heavy-tailed for all $j \in \{1, \ldots, D\}$. $\qquad\square$

### A.3 Notes on Assumption 1

Assumption 1 might look troublesome at first sight, but we will illustrate in this section that the condition is indeed very reasonable. We will show how to verify it in simple examples, and we will introduce a simpler, more intuitive sufficient condition for it.

First of all, let us present a restricted but more intuitive version of Assumption 1.

**Assumption 2** (Simplification of Assumption 1). *For all $j \in \{1, \ldots, D\}$ and $\lambda > 0$ it holds for $\mathbb{S} := [a,b]^{j-1}$ that there exist constants $z_j^*$ and $s > 0$ such that*

$$c\big(F_1(z_1), \ldots, F_j(z_j)\big) \geq se^{-(\lambda_\sigma-\varepsilon)z_j} \quad \text{for } z_j > z_j^* \text{ and } z_{<j} \in \mathbb{S} \ , \tag{9}$$

*where $c_j$ is the copula density of $q_{\leq j}$, $\lambda_\sigma$ is a lower bound of $\lambda\sigma(z_{<j})$, $\varepsilon > 0$ is small such that $\lambda_\sigma - \varepsilon > 0$.*

Let us summarize the simplifications that we make in Assumption 2. First of all, we restricted $\mathbb{S}$ to be a closed cube $[a,b]^{j-1}$, which is obviously a specific instant of a compact set with positive mass. Further, we assumed $\sigma$ to be continuous, and thus, $\lambda\sigma(z_{<j})$ must be lower-bounded in $\mathbb{S}$. This allows us to replace the function $f$ by the constant $\lambda_\sigma - \varepsilon$ for arbitrary small $\varepsilon > 0$.

After giving this simplified sufficient condition, we provide some intuition by presenting some examples where Assumption 2 holds true.

**Example 1** (Independent Variables). *Consider a random variable $\mathbf{z}$ with independent components, i.e. $q(z) = \prod_{j=1}^D q_j(z_j)$. Then, the associated copula is the independence copula (Figure 4a), which is a uniform random distribution on $[0,1]^D$. Therefore it is $c\big(F_1(z_1), \ldots, F_D(z_D)\big) = 1$ for all $z \in \mathbb{R}^D$ and Assumption 2 follows immediately since $s\exp(-(\lambda_\sigma - \varepsilon)z_j) \to 0$ for $z_j \to \infty$.*

**Example 2** (Bounded Copula Density). *Consider a lower-bounded copula density, i.e. there exists a lower bound $a > 0$ such that*

$$c(u_1, \ldots, u_D) \geq a \quad \text{for all } u \in [0,1]^D \ .$$

*Again, the validity of Assumption 2 in this setting is clear.*

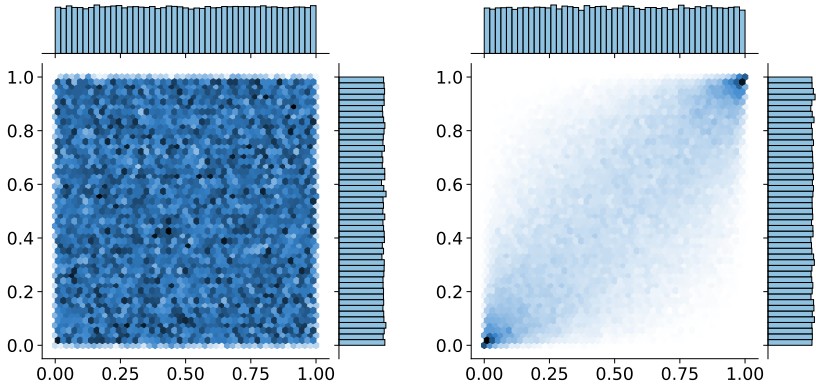

(a) 2-dimensional Independence copula: $c(z_1, z_2) = 1$.

(b) 2-dimensional Gaussian copula with correlation $\rho_{12} = 0.7$ .

Furthermore, this assumption is obviously not limited to bounded copula densities but also holds for copula densities that converge to 0 but whose decay rate in $z_j$ is lower-bounded by (9). To visualize the intuition, consider the 2-dimensional copula density of a Gaussian copula in Figure 4b. Imagine fixing $\mathbb{S}$ such that $F_1(z_1) \in [0.5, 0.75]$, which is compact for continuous $F_1$. Then (9) bounds the decay rate within the "tube" $[0.5, 0.75]$ if we consider $z_2 \to \infty$, i.e. if $F_2(z_2) \to 1$. Next, we show how we can formally prove the assumption for Gaussian copulae.

**Example 3** (Gaussian Copula). *The Gaussian copula with correlation matrix $R \in \mathbb{R}^{D \times D}$ has density function*

$$c(u) = \frac{1}{\sqrt{\det R}} \exp\left( -\frac{1}{2} \big( \Phi^{-1}(u_1), \ldots \Phi^{-1}(u_D) \big) \big( R^{-1} - I \big) \big( \Phi^{-1}(u_1), \ldots \Phi^{-1}(u_D) \big)^{\top} \right) ,$$

(10)

*where $I \in \mathbb{R}^{D \times D}$ is the identity matrix, $\Phi^{-1}$ is the inverse CDF of the univariate standard Gaussian distribution, and $u \in [0, 1]^D$. In the following, we consider Assumption 2 for $j = D$.*

*Note that $\mathbb{S}$ is assumed to be a compact set, therefore $F_j(z_j)$ are all upper and lower-bounded by some value for $z_{<j} \in \mathbb{S}$. This makes all polynomials of them also bounded. Hence, we can find constants $a', b', c'$ such that we can lower-bound the term within the exponential in (10) by*

$$-\frac{1}{2} \big( a' \Phi^{-1}\big(F_D(z_D)\big)^2 + b' \Phi^{-1}\big(F_D(z_D)\big) + c' .$$

*Plugging the above into (10) gives*

$$c\big(F_1(z_1), \ldots, F_D(z_D)\big) \geq \frac{1}{\sqrt{\det R}} \exp\left( -\frac{1}{2} \big( a' \Phi^{-1}\big(F_D(z_D)\big)^2 + b' \Phi^{-1}\big(F_D(z_D)\big) + c' \big) \right)$$

$$\propto \exp\left( -a \Phi^{-1}\big(F_D(z_D)\big)^2 + b \Phi^{-1}\big(F_D(z_D)\big) \right) \quad \text{(for some } a, b\text{)}$$

$$\geq \exp\left( -|a| \Phi^{-1}\big(F_D(z_D)\big)^2 + |b| \Phi^{-1}\big(F_D(z_D)\big) \right)$$

$$\geq \exp\left( -|a| \Phi^{-1}\big(F_D(z_D)\big)^2 \right) ,$$

(11)

*where the last line applies if $\Phi^{-1}\big(F_D(z_D)\big) \geq 0$, which is satisfied if $z^*$ is large enough[10].*

*Next, we use the asymptotic relation from Lemma 3*

$$\Phi^{-1}\big(F_D(z_D)\big) \sim \sqrt{-2 \log\big(1 - F_D(z_D)\big)} .$$

---

[10] for instance if $z^*$ is larger than the median of $\mathbf{z}_D$

*Hence, for each $\varepsilon > 0$ there exists a $z^*$ large enough such that*

$$\left| \frac{\Phi^{-1}\big(F_D(z_D)\big)}{\sqrt{-2\log\big(1 - F_D(z_D)\big)}} - 1 \right| < \varepsilon \ ,$$

*which can be rearranged to*

$$\Phi^{-1}\big(F_D(z_D)\big) < \sqrt{-2\log\big(1 - F_D(z_D)\big)}(1 + \varepsilon) \ .$$

*Plugging the above into* (11)*, we obtain*

$$c\big(F_1(z_1), \ldots, F_D(z_D)\big) \geq \exp\bigg(2|a|(1 + \varepsilon)^2 \log\big(1 - F_D(z_D)\big)\bigg)$$

$$= \big(1 - F_D(z_D)\big)^{2\tilde{a}} \ , \tag{12}$$

*where we define $\tilde{a} := 2|a|(1 + \varepsilon)^2$. Hence, we are left to lower-bound* (12)*, which we can do for a range of heavy-tailed marginal distributions such as:*

1. ***Pareto distribution:*** *The Pareto distribution with shape parameter $\alpha$ has CDF*

$$F(z) = 1 - \frac{1}{z}^{\alpha} \ .$$

   *Therefore,*

$$\big(1 - F_D(z_D)\big)^{2\tilde{a}} = \frac{1}{z_D}^{2|a|\alpha}$$

$$= \exp\big(-2\tilde{a}\alpha \log(z_D)\big)$$

$$\geq \exp\big(-2\tilde{a}\alpha z_D\big)$$

   *for $z_D \geq e$.*

2. ***Scale invariant distributions:*** *Following the same argument as above, each distribution with CDF*

$$F(z) = 1 - bz^{-\alpha} \quad \text{for } z > z^* \tag{13}$$

   *for constants $b, \alpha, z^* > 0$ satisfies the bound*

$$\big(1 - F_D(z_D)\big)^{2\tilde{a}} \geq b \exp\big(-2\tilde{a}\alpha z_D\big) \ .$$

   *Each distribution with CDF as in* (13) *is a scale-invariant distribution (see e.g. Theorem 2.1 in Nair et al. (2013)).*

3. ***Exponentially decaying distributions:*** *Every distribution that satisfies the bound*

$$F(z) \leq 1 - \exp(z)^{-\alpha}$$

   *for $z > z^*$ and $\alpha > 0$. In this case, we can again show that Assumption 1 is valid:*

$$\big(1 - F_D(z_D)\big)^{2\tilde{a}} \geq \exp(-2\tilde{a}\alpha z_D) \ .$$

Lastly, we want to emphasize that Corollary 1 is derived by an iterative application of Theorem 3. Therefore, Assumption 1 must hold for all "flow steps", i.e. if $T = T^{(L)} \circ \cdots \circ T^{(1)}$, we need to ensure validity of Assumption 1 for $\mathbf{z}^{(0)} := \mathbf{z}$, $\mathbf{z}^{(1)} := T^{(1)}(\mathbf{z})$, $\mathbf{z}^{(2)} := T^{(2)} \circ T^{(1)}(\mathbf{z})$, ..., $\mathbf{z}^{(L-1)} := T^{(L-1)} \circ \cdots \circ T^{(1)}(\mathbf{z})$. In Example 1, we show that this assumption holds true for $\mathbf{z}^{(0)}$ since we define our base distribution under the mean-field assumption. Furthermore, we conjecture that if we apply a Lipschitz-continuous diffeomorphism on a random variable with bounded copula density, then the transformed random variable must also have a bounded copula density. Hence, Assumption 1 would be valid for all "flow steps" (see Example 2) when starting with a mean-field base distribution $q(z) = \prod_{j=1}^{D} q_j(z_j)$. However, this conjecture needs to be studied in further research.

# B  ALGORITHMS AND COMPUTATIONAL DETAILS

## B.1  TAIL ESTIMATION

Many heavy-tailed distributions can be characterized by their tail index, which include the set of regularly varying distributions,[11] such as the $t$-distribution, the Pareto distribution, and many more. However, as already shown in Section 2.1, the tail index does not depend on the body of the distribution, and hence, non-tail samples must typically be discarded for tail index estimation. Although a variety of estimators for the tail index exist, such as the Hill estimator (Hill, 1975), the moment estimator (Dekkers et al., 1989), and kernel-based estimators (Csorgo et al., 1985), none of them is considered to be as superior in all settings. A major issue of all mentioned estimators is that they are based on a threshold defining the tail, i.e. the user needs to input statements of the form "the $k$ largest samples are considered to be tail events". Even though there exist some strategies to find $k$, there is none working robustly in all settings. In fact, one can construct simple counter examples for all estimators that lead to failures of tail estimation. This is due to undesired properties of the estimators, such as the lack of translation invariance of the Hills estimator (while the tail index clearly is location invariant). We refer to Section 9 in Nair et al. (2013) for a detailed text book treatment of tail index estimation. In summary, robust tail estimation is still considered as an unsolved problem, which forces practitioners to consider multiple estimators to make a well-founded decision. Furthermore, we note that the Hills estimator can only be applied for regularly varying distributions, which excludes the application of the Hills estimator to classify light-tailed distributions. In contrast, the moments and the kernel estimator can both be applied to identify heavy-tailed marginals and to assess a tail index.

To implement the tail assessment scheme, see Step 1 of the proposed method in Section 3.2, we found that Algorithm 1 works fine in classifying the correct tail behavior and giving a decent initialization for the tail indices. We reused the code by Voitalov et al. (2019), which implements all tail estimation procedures[12] from our Algorithm. Notice that we clip the tail index by 20, i.e. the algorithm classifies marginals with a tail index larger than 20 as light-tailed, which prevents a too restrictive set of allowed permutations, see Step 3 in Section 3.2. For illustration, consider the following simple example. Assume that we estimate all except of one marginal to be heavy-tailed. Then, the first component of the flow is never allowed to permute with other components, since they are classified as heavy-tailed. Hence, the mixing of the first component would be severely restricted. Further, since large tail indices indicate a less heavy-tailed distribution, it is reasonable to clip the tail index at some threshold.

---

**Algorithm 1** Marginal Tail Estimation

---

**Require:** Data_val
    tail.est $\leftarrow [\,]$
    **for** j in $\{1, \ldots, D\}$ **do**
        marginal $\leftarrow$ Data_val[:, j]
        moments $\leftarrow$ moments_est[|marginal|]        ▷ 0 if |marginal| is estimated to be light-tailed
        kernel $\leftarrow$ kernel_est[|marginal|]          ▷ 0 if |marginal| is estimated to be light-tailed
        **if** moments==kernel==0 **then**
            tail_est.append(0)    ▷ light-tailed if moments and kernel estimate a light-tailed marginal
        **else**
            hill $\leftarrow$ hills_est(|marginal|)
            **if** hill$> 20$ **then**
                tail_est.append(0)        ▷ light-tailed if hills estimator predicts high tail index
            **else**
                tail_est.append(hill)
            **end if**
        **end if**
    **end for**
    **return** tail_est

---

[11] see Section 2 in Nair et al. (2013) for further details

[12] including the hyperparameter selection (Danielsson et al., 2001; Qi, 2008; Draisma et al., 1999; Groeneboom et al., 2003)

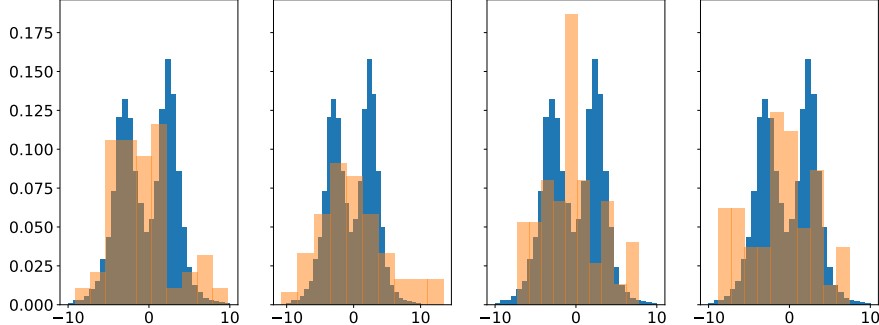

Figure 5: Applying the sample generation procedure from Section 4.2 but with tail samples from other base marginals and visualizing the 16th target marginal. Samples from the true distribution are visualized in blue and flow samples are in orange. We sample from the 3rd, 7th, 11th, and the 15th base marginal, respectively (from left to right). The visualized setting is $\nu = 2$ and $h = 8$.

## B.2 SYNTHETIC DATA GENERATION

Our generation of the synthetic distribution consists of 3 steps: 1. Generating the marginal distributions, 2. Defining a copula distribution, 3. Combining the marginal and the copula to obtain a multivariate joint distribution.

**Generating the marginal distributions.** The first four marginals are defined to be Gaussians. The following marginals are two 2-mixtures of two Gaussians and two mixtures of three Gaussians. The last $h \in \{1, 2, 4, 8\}$ components are a mixture of two $t$-distributions and the remaining marginals are again mixtures of two Gaussians. All mixtures have equal weight for each mixture component and all means and standard-deviations are randomized. Means are constructed by uniformly sampling from $[-4, 4]$, whereas standard-deviations are sampled from $[1, 2]$.

**Defining a copula distribution.** Recall, a Gaussian copula (10) is parameterized by a correlation matrix $R$. To generate $R$, we randomly sample 16 different pairs $(i, j) \in \{1, \ldots, 16\}^2$ with $i \neq j$ and set the corresponding entry of the correlation matrix

$$R_{i,j} := 0.25 \ .$$

**Obtaining a joint distribution.** Lastly, we combine the marginals with the Gaussian copula using Sklar's Theorem 4. This gives us a multivariate distribution with specified and complex marginals with a dependency structure given by the copula, see Joe (2014) for more details on the induced dependencies.

To construct the training, test, and validation sets $15.000$, $5\,000$, and $5\,000$ samples from this distribution are sampled, respectively.

## B.3 SAMPLING MARGINALLY TAIL EVENTS

In the main text, we have seen that the proposed tail generation procedure is successful in generating samples that are marginally in the tail of some specified marginal (Figure 2). In Figure 5, we present the 16th marginal when repeating the same experiment but with marginal tail samples from other marginals $\mathbf{z}_j$ than the proposed $\mathbf{z}_{k^{-1}(16)}$.

## B.4 HYPERPARAMETERS AND EXPERIMENTAL DETAILS

In all our experiments, we constructed a model using $L = 5$ MAF flow layers, where each emerging neural networks contains 1 hidden layer and 200 hidden neurons. The tail index in TAF is initially set to $\hat{\nu} = 10$. As a permutation scheme, we employ random permutations in the vanilla flow and

TAF, and random permutations within the sets of light- and heavy-tailed marginals, respectively, in mTAF. In addition, we applied batch-normalization after each triangular transformation. For optimization, we use an Adam optimizer with a learning rate of $1e-5$ and a weight-decay of $1e-6$. The learning rates of the tail indices are set to $0.1$ for TAF and for mTAF. We train until $200$ epochs or use early stopping if training gives no improvement after $30$ epochs. We use a batch size of $256$.

Our code is built with *PyTorch* (Paszke et al., 2019) and we make extensive use of the package *nflows* (Durkan et al., 2020) to implement mTAF. All code is provided along the submission.

