# OpenReview forum: "Marginal Tail-Adaptive Normalizing Flows"
_ICLR.cc/2022/Conference — ICLR 2022 Submitted_

### Official Review · Reviewer_dggh · 2021-10-21

**Correctness:** 4
**Technical Novelty And Significance:** 2
**Empirical Novelty And Significance:** 2
**Recommendation:** 3
**Confidence:** 4

**Main Review:**

 Strengths:
- The addressed problem is important. Vanilla normalizing flows (such as other deep density estimators) are poor at tail-estimation, which limits their applicability to many problems in science and engineering.
- The paper is very well written and it serves as a good introduction to both normalizing flows and heavy-tailed distributions.
- The offered solution is technically sound.
- The theoretical analysis is sound and convincing.

Weaknesses:
- The main weakness of the paper is its very limited novelty. All theoretical analysis and methodological improvements are relatively minor modifications of the work in (Jaini, 2020). The present paper does not contain major new ideas.
- The proposed method is somewhat inelegant as it proposes the use of a separate off-the-shelf tail estimator prior to the flow training. I do agree that it could be the right approach in many applications, however it is a rather obvious idea, not really worth of a top conference publication.

**Summary Of The Paper:**

The paper is composed of two main parts: 1) A theoretical section where the authors prove that (Lipschitz) triangular normalizing flows cannot map either heavy- or light-tailed base distributions into target distributions with different tail indices for different marginals; 2) An algorithmic section where the authors introduce a new method for modeling distributions with different tail indices for different marginals.

Both the theoretical and the algorithmic parts are a straightforward extension of the analysis and methods introduced in the paper "Tail-adaptive flows" (Jaini, 2020).

**Summary Of The Review:**

I highly appreciate the clarity and technical soundness  of the paper. However, I cannot recommend acceptance given the very limited novelty.

---

> ### Comment · Reviewer_dggh · 2021-11-29
> **Acknowledgment**
>
> dear authors,
>
> I do appreciate your general response. However, I still think that this paper is not up to standard for a top conference at the moment. I will therefore not change my score.

---

### Official Review · Reviewer_EStc · 2021-10-24

**Correctness:** 4
**Technical Novelty And Significance:** 2
**Empirical Novelty And Significance:** 2
**Recommendation:** 3
**Confidence:** 4

**Main Review:**

**Strengths:** I thought the paper was interesting. NFs definitely do have their limitations despite their expressivity, and I don’t think the problem of generating distributions with a mixture of both heavy- and light-tailed marginals has been considered before. The paper provides a more general definition of heavy tailedness that extends existing work and uses it to construct their mTAF method.

**Weaknesses:** That being said, I think the paper still requires a significant amount of work in order to demonstrate the efficacy of mTAF.
- First, it’s not clear to me when you would run into situations where you want to generate distributions with mixed-tail behavior in the marginals. I understand that it would be desirable to generate distributions with heavy tails, but when do we encounter cases where we would like to do both? I think making this clear would definitely strengthen the paper, and could also guide some downstream evaluation tasks.
- The experiments were probably the weakest aspect of the paper. mTAF was only evaluated on a synthetic dataset of 16 dimensions, which seems too small (even for tabular datasets commonly used for evaluating NFs). Also, the evaluations conducted in the experiments did not clearly demonstrate the advantage of mTAF over existing methods. For example in Table 1, does mTAF capture both the light-tailed and heavy-tailed components better than TAF/the base method? (this is hard to tell with just a simple average). Additionally, it’s hard to tell the difference between mTAF and TAF in Figures 2 and 3. I think the paper would be much stronger if the authors could find some compelling use cases of the method beyond synthetic Gaussians, and demonstrate that mTAF both captures all marginals more faithfully (via likelihoods) and can generate samples properly in the tails.
- I also think a big limitation of the method is that mTAF essentially requires separating out the light-tailed marginals from the heavy-tailed marginals (the permutation step where such marginals are grouped into 2 categories). This seems particularly problematic as the real advantage of using NFs is to learn complicated dependencies between all dimensions of the data to best capture the overall density. This is also why I was asking whether there are real-world examples where such mixtures occur, and whether this kind of ``independence assumption’’ makes sense in these scenarios. It seems like mTAF is very restrictive, and I am wondering if maybe that is why it doesn’t significantly outperform TAF and the vanilla baseline.

**Questions:**
- I’m also curious if the method performs worse relative to conventional flows (e.g. MAF) when the distribution in question is only light-tailed or heavy-tailed. It seems like if the tail index estimator is correct, mTAF should return the correct “tail behavior” of each marginal and generate either a light-tailed or heavy-tailed distribution only. Is this the case? Or does mTAF do a worse job at modeling, say, the light-tailed components, etc?
- Additionally, the vanilla baseline has pretty high variance and sometimes seems to perform on par with TAF -- would the authors elaborate upon this point?

**Miscellaneous/minor typos:**
- “Allow [us] to” in Section 2.1
- “By out theory” in Section 5



**Summary Of The Paper:**

This paper introduces Marginally Tail-Adaptive Flows (mTAFs), which extend existing work on TAFs to better learn a generative model of heavy-tailed distributions. In particular, they propose a new type of normalizing flow (NF) that can learn marginals with mixed-tail behavior.

**Summary Of The Review:**

Although the paper extends an existing approach to learn generative models of distributions exhibiting mixed-tail behavior, the paper has a number of weaknesses: (1) it’s not clear when such mixed-tail behavior arises in the real-world; (2) the class of flows considered are quite restrictive (affine, coupled with a permutation that requires light-tailed and heavy-tailed marginals to be split into two consecutive blocks); and (3) the empirical results are lacking: they only provide experiments on datasets of dim=16.

---

> ### Author Response · Authors · 2021-11-22
> **Evaluating Performance on Marginals**
>
> We are very grateful for your valuable comments. Especially, we like the idea of investigating the performance of mTAF on specific marginals. However, since NFs provide an estimate of the joint distribution, we cannot extract marginals in a straightforward manner. Instead, computing the marginals requires integrating out many components, bringing us back to methods like Monte Carlo approximations, which are tedious and slow to compute, and again, just approximations prone to errors, which might accumulate in a heavy-tailed regime. In our paper, we tried to circumvent this issue, by conducting our presented synthetical analysis: As further discussed in our general comment (point 2), we gradually increase the amount of heavy-tailed components, leading to a performance drop of vanilla, and a performance increment of TAF, while mTAF remains superior in all settings. These observations allow us to conclude that the heavy-tailed marginal components seem to be responsible for the different behavior of the different NFs, even though we did not examine their marginal fit directly.

---

> > ### Comment · Reviewer_EStc · 2021-11-28
> > **reply to author response**
> >
> > Thank you for the response that you provided in the rebuttal -- I'm confirming that I've read its explanation as well as the other authors' reviews. While I still believe that the problem is interesting, I wasn't convinced that the paper in its current state did an adequate job outlining the significance of the approach. Specifically, the theoretical contributions could be significantly bolstered by improved empirical evaluations. So at this time, I cannot recommend this paper for acceptance.

---

### Official Review · Reviewer_ZEHf · 2021-11-02

**Correctness:** 4
**Technical Novelty And Significance:** 2
**Empirical Novelty And Significance:** 1
**Recommendation:** 3
**Confidence:** 4

**Main Review:**

The paper is  The problem of estimating the tail behavior

**Pros**: Modelling tail-phenomena (or rare events) in general is a challenging problem made even more difficult in higher dimensions due to the lack of any definition of heavy/light tails in higher dimension. The problem considered by the authors of capturing tail behavior by normalizing flows and limitations of the chosen architecture for flow layers imposes is an interesting and valid problem. The proposed solution that uses estimators to estimate the tail-coefficient for choice of base distribution is interesting and a nice addition to start the optimization of the $\nu$ at a favorable location and to be closer to whats needed. Overall, the paper is very well written and easy to follow. The paper develops the idea in a natural and easy to understand manner.

**Cons**:

1)  **Motivation**: I found that the problem under consideration was not properly motivated and this issue lingers throughout the paper essentially making the paper come across as just an extension of Jaini et.al 2020. For example, it is not clear from the paper why capturing tails in variational inference paradigms is of importance. It can be shown that the error in modelling a probability density can be bounded arbitrarily well by learning the density properly on a bounded subset of the support of the density. Thus, what are the drawbacks in the model if it is unable to capture the tail phenomena present in the problem?

The authors do make statements regarding the limitations of the push-forward density given the choice of base density and the transformation map. However, I believe a more thorough discussion about the implications of these results (both in general and particularly for normalizing flows) and some recipe or ideas to alleviate these problems will help the paper tremendously.

2) **Significance**: Another weakness I believe of the paper itself is the significance itself. The main result of the paper ie that of specifying and ensuring that marginal distributions have the correct tail coefficient is a direct extension of the work of Jaini et.al 2020. In some ways, the optimization problem presented in Jaini et.al already can encompasses the correct marginal tails by optimizing over the $\nu$ vector. This weakness of significance is further amplified by the lack of strong empirical results. In the present form, the experimental results come across more as proofs-of concept rather than proving strong empirical support. Furthermore, I'd be interested to see the gain in starting the optimization with tail-coefficients estimated using the various estimators vs random initialization and letting the process figure these out. Will the first step of using estimators lead to any significant gains?

**Other comments:**
- Definition 4 it seems is a bit restrictive as well since it completely side-steps any issues with differences in tail-coefficients. For examplke different ,marginals can be heavy tailed but have different degrees of heaviness. In that case, is it correct to call the two distributions having the same tail behavior?

- It seems in the experiments that if the tail estimator estimates that some marginals are light-tailed, a normal distribution is used. However, again there can be degrees of light-tailedness (see Jaini et.al 2020) eg. uniform vs normal. In these scenarios too, a lighter tailed distribution cannot be pushed-forward to another light-tailed distribution but with higher tail coefficient with lipschitz maps. Thus, the problem of mismatched tails may still persist.

**Summary Of The Paper:**

The paper proposes an extension to Tail-adaptive flows for learning the tail behavior of target distributions using normalizing flows. The authors propose to learn the tail behavior by learning flows that match the tail properties of the marginal distributions. They achieve this by using a source distribution consisting of marginal distributions with tail properties matching the target distribution. The tail coefficient of the source distribution is set in a data-driven manner using estimators that can estimate this tail coefficient.

**Summary Of The Review:**

Overall, the paper studies a pertinent and difficult problem. However, in the current form, the present manusript provides only initial proofs-of-concept for potentially interesting ideas. These ideas need to be demonstrated and explored in more detail both in theory and empirically to make the manuscript stronger.

---

> ### Author Response · Authors · 2021-11-22
> **Authors' Response**
>
> We appreciate your insightful and thorough review and want to take the chance to comment on some critique points.
> 1. *Extending VAEs.* Even though related works are quite scarce, there exist some in this context: [1] show that a heavy-tailed prior is indeed well-suited from an optimization point of view. The authors show that a heavy-tailed prior is able to robustify the training of the VAE. Similar things have also been observed for NFs [2]. However, it is still an open question, as far as we know, whether VAEs are able to model heavy-tailed distributions (or whether one could enhance the performance by modifying the prior), which might be an interesting direction for future works.
> 2. *Drawbacks of a model that does not learn the tails*. Of course, there exist applications where tail-events have a massive effect. These include applications in climate catastrophes and insurances or finance. Since a generative model might be a crucial part of a machine learning system, a wrong evaluation of the tail-likelihoods can have a large effect: For these systems, it might have a significant effect whether an event is assigned a probability of 1e-3 or 1e-6.
> 3. *Different degrees of heavy- and light-tailedness*. As correctly pointed out by the reviewer, there are different forms of light-tailedness. As it has been shown by [3], it is impossible to modify the normalizing flow to account for distributions that have a support with different topological properties, such as distributions with bounded support. In this work, we restricted ourselves to the simple setting where the true density p is positive, that is, we consider unbounded support. On the other hand, note that [3] does only cover the cases where base and target distributions have nonhomeomorphic supports, which might not be the case in our investigated settings. Instead, we distinguish distributions via their tailedness, which is not directly related to their support (t-distributions and normal distribution have the same support for instance). Hence, [3] covers a somewhat different direction of limited expressiveness of NFs. It would indeed be an interesting direction to combine both orthogonal ideas, which is left for future research.
>
>
> [1] Student-t Variational Autoencoder for Robust Density Estimation, Takahashi, H., et al., 2018
> [2] Robust model training and generalisation with Studentising flows, Alexanderson, S., Eje Henter, G., 2020
> [3] Relaxing Bijectivity Constraints with Continuously Indexed Normalising Flows, Cornish, R., et al., 2021

---

### Official Review · Reviewer_pWUG · 2021-11-03

**Correctness:** 3
**Technical Novelty And Significance:** 3
**Empirical Novelty And Significance:** 2
**Recommendation:** 3
**Confidence:** 3

**Details Of Ethics Concerns:**

I did not have ethics concerns

**Main Review:**

Strength:

1. Solid and rigorous mathematics foundation

The theoretical proofs are very helpful and provide a clear insight, which explains the motivation and intuition.

2. Careful discussion about the related work and current limitations

The authors provide a good review and comparison of the existing works. Also, the limitations and future directions are helpful and insightful.

3. Well-written and easy to follow

The paper is well-structured and easy to follow. The theoretical proof strongly connects with the experiments, which makes the papers much easier to understand.

Weakness

 1. The novel contribution is marginal

This work is mainly inspired by Jaini et al, 2020, who proposed to model long-tailed distribution via normalizing flows. Although the theoretical contribution is strong, the new proposed mTAFs did not show a significant improvement, as shown in Table 1, compared with vanilla and TAF.

2. Lack of baseline methods and comparison

The SOTA flow models and architectures are not included in the baselines.  Although it is argued by the authors in the potential work, I still believe the comparison is necessary. Many papers have shown that the affine coupling layers, such as RealNVP, have limited expressivity in handling complex distribution. Either light-tailed or heave-tailed distribution would be more challenging. So the worse performance might be due to the limited representation capability of the vanilla flows.

3. Experiments need to be improved with large-scale and high-dimensional datasets

Currently, only synthetic toy examples are provided to demonstrate the performance. The dimensionality is also low. If the proposed algorithm is able to scale to high-dimensional problems, it would be very helpful to increase the impact.


**Summary Of The Paper:**

This paper focuses on understanding the tail behavior of normalizing flows through a mathematical and statistical way. Motivated by Jaini et al 2020's work on learning long-tailed distribution via triangular flows, this work proves that the marginal tailedness can be controlled by the tailedness of the marginals of the base distribution in flow-based models. Based on this theoretical insight, the authors propose a new algorithm by leveraging a data-driven permutation scheme to enable a correct tail behavior of the target distribution.



**Summary Of The Review:**

The paper provides a strong theoretical insight but the experiments and baselines are weak.  The contribution seems limited with a marginal improvement compared with the current work.

---

### Author Response · Authors · 2021-11-22
**General Comments**

Thank you very much for your reviews! We are glad that the reviewers appreciate the relevance of the investigated problem and the derived theory. The valuable comments and suggestions will help us in improving the quality of the paper further.
Here, we will discuss the points that a majority of the reviews have in common.

1. *Mixed-tailed distributions in practice.*
We should have put more emphasis on explaining the existence and relevance of mixed-tailed distributions, which were the focus of this work, in practice. Let us elaborate a bit more on situations in which such distributions emerge. In fact, one can always consider two, let's say 1-dimensional distributions, and study the joint distribution to investigate the interactions between both components. For instance, let us consider a generative model for some insurance application, where we construct the joint distribution of i) insurance losses and ii) the age of the insurant. It is obvious, that this joint distribution has a heavy-tailed (insurance losses) and a light-tailed (age) marginal, and therefore, is a mixed-tailed distribution according to Definition 4. In addition, many functional relationships between marginal random variables induce heavy-tails, leading to mixed-tailed behaviors. Imagine we have some multivariate distribution, where the first marginal $x_1$ is standard normal distributed, and the second marginal has the following functional relationship to the first one: $x_2=\exp(x1)$. Then, $x_2$ is a log-normal distributed marginal, which is heavy-tailed. Another example might be the ratio of random variables, such as the ratio of two standard normal distributed random variables, which is a Cauchy distribution. Of course, these are just simple examples, which, however, can be generalized to more complex and realistic models.

2. *Synthetic experiments*.
Actually, we are a bit surprised that the reviewers do not find the improvements shown in Table 1 significant. First of all, and it seems that we did not succeed in explaining this well enough, we can observe a behavior of vanilla MAF and TAF that goes along the lines of our derived theory. When the number of heavy-tailed components increases, the performance of TAF increases. On the contrary, TAF shows worse performance, if only a few of the marginals are heavy-tailed. This can be explained by Proposition 2, which states that TAF learns exclusively marginally heavy tails. This is an important extension to the work by Jaini et al. since it shows crucial limitations of TAF, which we aim to address in this work. The vanilla MAF shows a reversed behavior, i.e. its performance decreases when we increase the number of heavy-tailed components. This tradeoff between vanilla MAFs and TAF provides us with useful and important insights, which help us to broaden our understanding of NFs.
In fact, vanilla MAF and TAF are recovered in the extreme cases: mTAF equals vanilla MAF  if we cannot find any heavy-tailed component in the data distribution, and it equals TAF (with D degrees of freedom) if all components are classified as heavy-tailed. Therefore, mTAF can be interpreted as an interpolation between vanilla MAF and TAF (with D degrees of freedom). This flexible model design leads to significantly higher log-likelihoods, where the significance has been tested using a t-test.

3. *Scaling up to real-world data.*
On this point, we agree with the reviewers: A more elaborate real-world data analysis, which we are currently working on, will improve the quality of this work remarkably. In a next version, we will scale up the experiments and include experiments, which combine SOTA NF architectures with our proposed modification.

4. *Limited Novelty.*
We are convinced that our analysis is a substantial extension to prior works since we a) pointed out limitations of the approach by Jaini et al. by extending the theoretical understanding of the relation between the tail behavior of the marginals of the base and the target distribution, and b) proposed a new method based on these insights, which we do not see as a trivial extension of previous work (i.e. TAF). This new extension requires us to derive a specific permutation scheme and, additionally, we need to employ techniques to estimate the tail index. As we have explained in 1., we want to emphasize that many real-world data sets are indeed mixed-tailed, and hence require a different treatment, see the discussion in 2. These points highlight the importance of mTAF, which is not a minor extension but should rather be seen as a generalized tool that can pick the best of both models, vanilla MAF and TAF.

---

### Decision · Program_Chairs · 2022-01-20

**Decision:**

Reject

**Comment:**

This paper addresses the performance of normalizing flows in the tail of the distribution. It does this by controlling tail properties in the marginals of the high-dimensional distribution. The paper is well-motivated, and the key theoretical insight has merit. However, the general perspective and methodology appears to be incremental relative to past results. Furthermore, some concerns over correctness remain after discussion with authors. Also, clear baselines and more realistic settings are lacking in the experimental results. Thus, while the paper generally has promising ideas on a pertinent topic, it appears to be not developed enough to merit dissemination.